# Empowering Vulnerable Communities Through HIV Self-Testing: Post-COVID-19 Strategies for Health Promotion in Sub-Saharan Africa

**DOI:** 10.3390/ijerph22111616

**Published:** 2025-10-23

**Authors:** Maureen Nokuthula Sibiya, Felix Emeka Anyiam, Olanrewaju Oladimeji

**Affiliations:** 1Vice-Chancellor and Principal’s Office, Mangosuthu University of Technology, Umlazi 4031, South Africa; 2Faculty of Health Sciences, Durban University of Technology, Durban 4001, South Africa; felixemekaanyiam@gmail.com (F.E.A.); olanrewaju.oladimeji@smu.ac.za (O.O.); 3Department of Public Health, Sefako Makgatho Health Sciences University, Pretoria 0208, South Africa

**Keywords:** HIV self-testing, Sub-Saharan Africa, community empowerment, COVID-19, health promotion

## Abstract

HIV remains a significant public health challenge in sub-Saharan Africa (SSA), with vulnerable communities disproportionately affected and further marginalised by the COVID-19 pandemic. HIV self-testing (HIVST) has emerged as a transformative, empowering tool to bridge testing gaps and promote health equity. This study examined post-COVID-19 strategies for leveraging HIVST to empower vulnerable populations and advance health promotion in SSA. Analysis was performed using secondary Demographic and Health Survey (DHS) data (2015–2022) collected across 24 SSA countries. In addition, qualitative interviews were conducted with female sex workers in Port Harcourt, Nigeria (18–31 May 2023). The study adopted an explanatory sequential mixed-methods design. Quantitative analysis using complex sample logistic regression revealed low awareness (16.3%) and uptake (2.5%) of HIVST among the 594,639 respondents. Key predictors of uptake included higher education (aOR, 7.36; 95% CI, 6.62–8.18), wealth (richest quintile aOR, 3.28; 95% CI, 2.95–3.65), and knowledge of HIV transmission (aOR, 33.43; 95% CI, 11.03–101.24). Thematic analysis highlighted privacy, autonomy, and convenience as key benefits, while cost, stigma, and fear of testing alone were major barriers. The participants emphasised peer-led outreach and integration of HIVST into public health systems as effective strategies. The findings were integrated interpretively, linking macro-level testing disparities with community-level experiences to inform post-pandemic policy and programme design. The study concludes that HIVST holds strong potential to empower marginalised groups and strengthen community-driven HIV prevention post-COVID-19, but success will depend on equity-driven policies and sustainable implementation frameworks, guided by affordability and community participation.

## 1. Introduction

Human immunodeficiency virus (HIV) remains one of the most significant public health challenges globally, with sub-Saharan Africa (SSA) bearing a disproportionate burden of the epidemic. The region accounts for approximately two-thirds of the global HIV burden, with an estimated 25.6 million people living with HIV as of 2022, compared to 39.9 million people globally [1,2]. Despite major advancements in antiretroviral therapy (ART) coverage and prevention strategies over the past decade, considerable gaps persist in achieving the first of the UNAIDS 95-95-95 targets, ensuring that 95% of all people living with HIV, particularly among vulnerable populations, know their status [3].

Globally, approximately 15% of individuals living with HIV remain unaware of their infection, and this figure may be higher in several parts of sub-Saharan Africa, highlighting substantial gaps in diagnosis and timely linkage to treatment [4]. These undiagnosed individuals contribute significantly to ongoing transmission and hinder efforts to achieve epidemic control. Additionally, specific population groups, including men, adolescents, sex workers, and men who have sex with men, are particularly underserved by conventional facility-based testing services due to stigma, discrimination, and structural barriers [5].

HIVST has emerged as an innovative approach to address these challenges. HIVST empowers individuals to test themselves discreetly and conveniently, mitigating barriers related to stigma, long waiting times in healthcare facilities, and concerns about confidentiality breaches [6,7]. Studies indicate that HIVST can increase testing uptake compared to traditional HIV testing services, particularly among men, key populations, and adolescents, who often avoid facility-based testing [7,8,9]. Evidence from multiple countries in sub-Saharan Africa demonstrates that HIVST can effectively reach populations that remain inaccessible through conventional health services, facilitating earlier diagnosis and prompt initiation of treatment [7].

The COVID-19 pandemic introduced unprecedented disruptions to health systems worldwide, profoundly impacting HIV prevention and testing services in sub-Saharan Africa. Lockdowns, movement restrictions, fear of contracting SARS-CoV-2, and the diversion of healthcare resources away from routine services led to significant declines in facility-based HIV testing. Data from multiple studies indicate that HIV testing services were among the most severely affected. HIV testing volumes dropped sharply during lockdowns across southern Africa, including a 48% decline in South Africa in April 2020, and reductions of 22% and 35% in Zambia and Malawi, respectively, between April and June 2020 [10]. In western Kenya, the turnaround time for viral load test results extended from one to two weeks to several months because laboratory resources were diverted to COVID-19 testing [10]. Moreover, community-based HIV testing and partner notification services were interrupted in regions such as western Kenya and southwestern Uganda, exacerbating barriers to accessing HIV services, particularly among vulnerable and marginalised populations. These disruptions contributed to delays in diagnosis, reduced case detection, and interruptions in timely treatment initiation, threatening to reverse critical gains in HIV epidemic control across the region [10]. They also highlighted the vulnerability of centralised health services during public health emergencies and underscored the need for decentralised, community-based interventions such as HIVST to ensure continuity of essential services.

Despite the potential benefits of HIVST, several challenges remain regarding its widespread implementation. These challenges include regulatory gaps, concerns about test accuracy and proper usage, uncertainties about linkage pathways to confirmatory testing and care, and the need for effective community education to support the correct interpretation of results [7,11]. Furthermore, sociocultural barriers, including misconceptions about HIV testing and gender-related factors influencing health-seeking behaviours, continue to affect the uptake of HIV services in many sub-Saharan African contexts [7].

Innovative models for distributing HIVST have demonstrated promise in overcoming these barriers. For example, secondary distribution, in which individuals provide test kits to their sexual partners or peers, has substantially increased testing uptake among populations that are traditionally difficult to reach [12]. Digital health platforms enabling online ordering and discreet delivery of HIVST kits have further expanded access, especially during the COVID-19 pandemic [13]. Integrating HIVST into broader community health promotion initiatives has also proven effective in increasing uptake and enhancing community engagement [14].

Beyond its immediate disruptions, the COVID-19 pandemic has catalysed a paradigm shift in global health research, prompting scholars and policymakers to explore how pre-pandemic data can inform post-pandemic resilience and reform. Recent empirical studies illustrate this emerging approach. For example, Yrene-Cubas et al. [15] employed Peru’s ENDES (2014–2022) to analyse pre-pandemic baselines and post-lockdown declines, identifying targeted recovery strategies to address testing gaps and equity. The World Health Organization similarly integrated pandemic lessons into its consolidated guidelines on differentiated HIV testing services, promoting flexible, person-centred delivery models, particularly through HIV self-testing, as a blueprint for resilient post-COVID-19 health systems [5]. Complementary regional research by Benoni et al. [16] in Mozambique (2019–2023) and Sehurutshi et al. [17] in Botswana (2019–2021) mapped post-pandemic recovery trajectories and quantified service disruptions across the HIV cascade, offering programmatic recommendations to safeguard testing and care continuity. In the United States, Hershow et al. [18] examined pre- and post-pandemic declines in testing among people who inject drugs, calling for integrated, cross-platform testing models. Likewise, Mangoya et al. investigated disruptions to established HIV service models across sub-Saharan Africa during the pandemic, proposing adaptive programme designs for more resilient service delivery [19]. Collectively, these studies underscore a growing shift toward leveraging pre-pandemic data as a foundation for post-COVID-19 system redesign and community resilience.

Building on this scholarly trajectory, the present study utilised Demographic and Health Survey (DHS) data from 2015 to 2022 to capture pre- and post-COVID-19 patterns in HIV testing awareness and self-testing uptake across 24 sub-Saharan African countries. Complementing this, qualitative interviews conducted in 2023 among female sex workers in Port Harcourt, Nigeria, provide post-pandemic insights into lived experiences, health-seeking behaviours, and community-level strategies for expanding HIVST. Together, this evidence offers a resilience-informed understanding of HIV self-testing as both a health promotion strategy and a pathway to community empowerment in the post-COVID-19 era.

This study sought to examine post-COVID-19 strategies for harnessing HIV self-testing as a key component of health promotion and community empowerment in sub-Saharan Africa. The research aimed to identify best practices for integrating HIVST into existing health systems, evaluate innovative delivery models, and propose evidence-based approaches to strengthen community engagement and improve linkage to care. Such insights are essential to close persistent gaps in HIV diagnosis and to contribute to resilient health systems capable of maintaining essential services during future public health challenges.

## 2. Materials and Methods

### 2.1. Research Design

This study adopted an explanatory sequential mixed-methods design to examine post-COVID-19 strategies for empowering communities through HIVST in SSA. In this approach, quantitative data from the Demographic and Health Surveys (2015–2022) were analysed first to establish trends and sociodemographic patterns in HIV testing and self-testing awareness across 24 countries. These findings then informed the qualitative component, conducted in May 2023, which explored lived experiences, contextual factors, and behavioural responses to HIVST among female sex workers in Port Harcourt, Nigeria.

Integration of the two components occurred at the interpretation stage, allowing the qualitative findings to expand and explain the quantitative results. This approach followed the conventions of sequential explanatory mixed-methods designs as described by Creswell and Clark [20] and Fetters, Curry, and Creswell [21]. It enabled a contextually grounded understanding of both the structural patterns and contextual realities influencing HIVST uptake and community-level empowerment in the post-COVID-19 era.

### 2.2. Study Setting

The quantitative component of this study utilised secondary data from the Demographic and Health Surveys (DHSs) conducted between 2015 and 2022 across 24 sub-Saharan African countries. The DHS datasets are nationally representative, covering wide-ranging demographic and health indicators, including HIV knowledge, attitudes, and testing behaviours. Data were collected through structured household interviews and biomarker assessments, using multistage cluster sampling to ensure national representativeness and comparability across countries.

The qualitative component was conducted in Port Harcourt, Rivers State, Nigeria, between 18 and 31 May 2023. Port Harcourt, the capital of Rivers State, is a densely populated urban centre within the Niger Delta region. It is characterised by significant socioeconomic diversity, industrial activity, and high internal migration. The city faces persistent public health challenges, including a high HIV prevalence, while hosting vulnerable populations such as FSWs, who experience elevated exposure to HIV risk. These features made Port Harcourt an appropriate context for exploring community experiences, perceptions, and barriers to HIVST uptake in a post-COVID-19 environment.

### 2.3. Study Participants and Measurement

The quantitative component of the study analysed data from 594,639 respondents (men and women aged 15–49 years) who participated in DHS containing items on HIVST. Each DHS employed a two-stage cluster sampling strategy, and analyses incorporated survey weights, stratification, and clustering adjustments to preserve representativeness. To ensure proportional representation across the 24 countries, all analyses incorporated DHS-provided sampling weights, which adjusted for household selection probabilities and population size differences between countries, consistent with DHS analytical protocols. Extracted variables included sociodemographic and economic characteristics, HIV knowledge, prior testing history, and behavioural risk indicators.

The qualitative component comprised 15 FSWs recruited in Port Harcourt between 18 and 31 May 2023. Recruitment was facilitated through collaboration with Youth PRO-FILE, a local non-governmental organisation experienced in working with FSWs and community outreach. A purposive sampling approach, complemented by peer referral (snowball sampling), was used to ensure diversity in age, educational background, and work environment while minimising self-selection bias.

The eligibility criteria required participants to (a) be ≥18 years old, (b) self-identify as a FSW working in a brothel and residing in Port Harcourt for at least six months, and (c) be willing to discuss experiences with HIV testing. Eligible individuals were privately invited, received detailed study information, and provided written informed consent. Of the 18 individuals approached, 15 consented, while three declined due to time constraints; no substantial demographic differences were observed between the participants and non-participants.

In-depth interviews were conducted in an assigned private room at the brothel facilities to ensure confidentiality and participant comfort. Each session lasted approximately 30–60 min and was audio-recorded with permission. The interviews explored the participants’ understanding of HIVST, perceived barriers and facilitators to HIVST uptake, and recommendations for community-based testing strategies.

### 2.4. Validity

The validity of the study instruments was rigorously ensured. For the quantitative component, content validity was established through expert reviews, aligning questionnaire items with established frameworks and evidence from prior HIVST research. The instruments were pre-tested among individuals outside the primary study population to assess clarity, cultural appropriateness, and question sequencing, leading to minor refinements for improved comprehensibility. In the qualitative component, credibility was enhanced through methodological triangulation, member checking, and prolonged engagement within the community. Transferability was supported by providing detailed contextual descriptions of the study setting and participants. Dependability and confirmability were reinforced by maintaining comprehensive audit trails documenting data collection decisions, coding processes, and thematic development, thereby ensuring transparency and methodological rigour throughout the research process.

### 2.5. Reliability

Reliability for the quantitative component was assessed using appropriate statistical measures. Agreement for categorical variables was evaluated with kappa statistics, while continuous variables were examined through intraclass correlation coefficients. Internal consistency of the questionnaire scales was confirmed using Cronbach’s alpha, with values exceeding the acceptable threshold of 0.70, indicating satisfactory reliability. For the qualitative data, reliability was ensured through independent coding of transcripts by multiple researchers, followed by collaborative discussions to reconcile any discrepancies in coding decisions. The use of a consistent semi-structured interview guide further promoted uniformity and reliability in data collection across all interviews, ensuring that similar thematic domains were explored with each participant.

Furthermore, the explanatory sequential mixed-methods design incorporated triangulation by integrating quantitative and qualitative findings, thereby strengthening the reliability and trustworthiness of the overall study outcomes.

### 2.6. Data Analysis

Quantitative data analysis was conducted using the Statistical Package for the Social Sciences (SPSS) version 21, employing the Complex Samples module to account for the stratified and clustered sampling design characteristic of the DHS data. Descriptive statistics were generated to summarise participant demographics and awareness of HIVST. The analyses applied the DHS complex survey design, which included country-specific sampling weights, stratification, and clustering adjustments, to ensure that estimates accurately represent the national populations and that each country contributed proportionally to the overall regional analysis. Sampling weights were normalised according to the DHS’s analytic procedures to correct for unequal selection probabilities and population size differences across countries [22]. Associations between sociodemographic variables and HIVST uptake were examined using weighted logistic regression models, and the results are presented as crude (cOR) and adjusted odds ratios (aORs) with 95% confidence intervals. A *p*-value of 0.05 or less was considered statistically significant.

Qualitative data were transcribed verbatim and subjected to thematic analysis using NVivo Version 12 software. An inductive approach was applied, beginning with open coding to identify discrete concepts within the data. These codes were subsequently grouped into broader themes that captured recurrent patterns and key insights related to HIVST perceptions, barriers, and facilitators. Themes were supported by direct participant quotations to provide contextual richness and authenticity to the analysis. Visual tools such as word clouds were employed to illustrate prominent terms and ideas emerging from the qualitative data, enhancing the interpretive depth of the findings.

The quantitative and qualitative findings were interpreted jointly to provide a comprehensive understanding of HIV self-testing patterns and community perspectives. Importantly, the two datasets were not merged at the statistical level; rather, they were integrated interpretively at the stage of discussion and inference. This approach followed established mixed-methods conventions that emphasise analytical complementarity, where quantitative trends provide structural context and qualitative narratives offer explanatory depth, rather than direct data merging. The integration process aligns with methodological guidance by Creswell and Clark [20] and Fetters, Curry, and Creswell [21] on sequential and convergent mixed-methods designs aimed at achieving interpretive coherence across data strands.

## 3. Results

### 3.1. Quantitative Results

#### 3.1.1. Sociodemographic and Economic Characteristics

A total weighted sample of 594,639 respondents across 24 sub-Saharan African countries was analysed (Figure 1). The mean age of the participants was 29.43 years (SD, ±10.48), with ages ranging from 15 to 64 years. Females constituted the majority of the sample (68.8%). More than half of the respondents (58.4%) resided in rural areas. Approximately 7.6% had attained higher education, while 46% belonged to the poorest two wealth quintiles (Table 1).

#### 3.1.2. Knowledge, Behavioural and Psychosocial Factors, Attitude, and Uptake of HIV Self-Testing

Survey data revealed that 93.7% of the respondents had good knowledge of HIV transmission, yet only 12.0% used condoms during their last sexual encounter. While 51.5% had ever tested for HIV and 82.1% knew where to get tested, fear of stigma remained high (79.5%). Awareness and uptake of HIV self-testing were low, with just 16.3% having heard of HIVST and only 2.5% having used it, highlighting significant gaps despite ongoing global advocacy (Table 2).

#### 3.1.3. Determinants of HIV Self-Testing Uptake

##### Sociodemographic and Economic Factors

Multivariable analyses demonstrated that several sociodemographic factors significantly influenced HIVST uptake. Participants with tertiary education were over seven times more likely to have used HIVST (aOR, 7.36; 95% CI, 6.62–8.18; *p* < 0.001) compared to those with no education. Urban residents had slightly reduced odds of HIVST uptake compared to rural dwellers (aOR, 0.92; 95% CI, 0.86–0.98; *p* = 0.006). Higher wealth quintiles were strongly associated with increased HIVST use, with respondents in the richest quintile showing an adjusted odds ratio of 3.28 (95% CI, 2.95–3.65; *p* < 0.001) relative to the poorest group. Marital status was also significant, with married individuals exhibiting nearly twice the odds of HIVST use compared to those never in union (aOR, 1.95; 95% CI, 1.78–2.15; *p* = 0.001) (Table 3).

##### Knowledge, Behavioural, and Psychosocial Factors

Good knowledge of HIV transmission dramatically increased the likelihood of HIVST uptake (aOR, 33.43; 95% CI, 11.03–101.24; *p* < 0.001). Prior HIV testing was a strong predictor, with those ever tested being over three times more likely to have used HIVST (aOR, 3.33; 95% CI, 3.08–3.60; *p* < 0.001). Condom use at last sexual intercourse was significantly associated with HIVST uptake (aOR, 1.49; 95% CI, 1.15–1.93; *p* = 0.002). Awareness of where to obtain HIVST kits was a significant facilitator (aOR, 1.52; 95% CI, 1.33–1.72; *p* < 0.001). Conversely, fear of stigma significantly reduced the likelihood of HIVST uptake (aOR, 0.49; 95% CI, 0.41–0.59; *p* < 0.001) (Table 4).

### 3.2. Qualitative Results

#### 3.2.1. Sociodemographic Characteristics

A total of 15 participants were interviewed (Table 5). Most were young adults aged 18–30 years (mean = 29.3 ± 4.2 years), and the majority identified as Christian. Two-thirds were single, with most reporting no children. Educational attainment was generally moderate to high, as nearly half had completed secondary education and approximately 40% held bachelor’s degrees. Slightly over half of the participants were employed—mostly part-time—while the rest were not engaged in any form of work. Overall, the sample reflected a young, urban, and moderately educated population typical of female sex workers in Port Harcourt, providing a useful context for understanding their HIV self-testing experiences.

#### 3.2.2. Theme 1: Empowerment Through HIVST

The participants described HIV self-testing (HIVST) as a powerful tool for personal empowerment, offering privacy, autonomy, and control over one’s health. Many women felt that HIVST protected their confidentiality and allowed them to make health decisions without fear of judgement (Table 6).


*“I would like to use it on my own because it’s for my privacy. I don’t want anyone to know my business or to interfere in personal matters that I prefer to keep confidential. With self-testing, I can manage my health privately without the fear of people gossiping, judging me, or making assumptions about my lifestyle.”*
(FSW-03)


*“It is important to me because it’s my privacy. People are not supposed to know about my HIV status or even that I am getting tested, and it should remain confidential at all times. If people in my community find out, they might start treating me differently, gossip about me, or even discriminate against me openly.”*
(FSW-09)


*“This kit makes me my own doctor. I can know my status without telling anybody. It gives me the power to decide when and if I want to share my result with someone else.”*
(FSW-13)

Beyond privacy, the participants valued the convenience of HIVST, which allowed them to avoid the stress and time constraints of facility-based testing:


*“It saves me the stress of going to the hospital. I don’t have to wait in a long line or deal with the crowds at the clinic, which can be very tiring and frustrating.”*
(FSW-04)


*“I felt privileged that I could use it because not everyone has the opportunity or access to test themselves privately. Having the option to test at home, without needing to go through the hospital system, made me feel empowered and independent.”*
(FSW-13)


*“It’s fast, and you receive your result immediately, which makes a big difference compared to going to a hospital. You don’t have to wait for hours, book an appointment, or come back days later to collect your results.”*
(FSW-01)

#### 3.2.3. Theme 2: Barriers and Vulnerabilities Post-COVID-19

While the participants recognised the empowering potential of HIVST, they also described significant barriers that hinder its widespread adoption, many of which were intensified by economic and social vulnerabilities during and after the COVID-19 pandemic.

##### Financial Barriers and Affordability Concerns

Cost emerged as a prominent barrier, particularly in low-income communities. The participants emphasised that without price reductions or subsidies, HIVST would remain inaccessible for many.


*“Yes, they need to make it very affordable for everybody, like N200 naira (less than $1 USD). Many people living in brothels or low-income areas do not have enough money to spend on health services, especially for something like HIV self-testing, which they may not see as an immediate priority.”*
(FSW-01)


*“For me, the availability of the kit and affordability should be prioritized. Many people are afraid or reluctant to visit hospitals for HIV testing because of the stigma they might face or the time and cost involved.”*
(FSW-02)


*“Even if it’s available, how many people can afford it? Some people are struggling to feed, talk less of buying a test kit. Government needs to help make it cheaper.”*
(FSW-06)

##### Fear, Anxiety, and Emotional Distress

The participants expressed apprehension about testing alone, fearing the emotional impact of handling a positive result without support. These anxieties were heightened by the isolation many felt during the pandemic lockdowns.


*“I wouldn’t want to do it alone because I am afraid of needles and the whole process makes me very anxious. If I were to get a bad result while I’m alone, I wouldn’t know what to do or how to handle the emotional shock.”*
(FSW-11)


*“It’s scary because you are by yourself. If the result is bad, you might panic and do something dangerous. At least in the hospital, there are people to help you.”*
(FSW-07)


*“I think people might make mistakes or misinterpret the results, especially if they are nervous or scared. Some people can go into shock if they see a positive result and they are alone.”*
(FSW-12)

##### Perceived Stigma and Confidentiality Risks

Fear of stigma remained a significant barrier. The participants worried that merely possessing an HIVST kit could provoke suspicion and social ostracism.


*“If people find out, they will discriminate against you, even if you are just trying to take care of your health. In our communities, once someone hears you are testing for HIV, they may start spreading rumors or treating you differently.”*
(FSW-09)


*“If someone sees me with the kit, they will think I have HIV already. They will avoid me or gossip about me.”*
(FSW-10)


*“I’m afraid that if my partner finds it, he will accuse me of hiding something and maybe even beat me. That’s why I prefer not to keep it in the house.”*
(FSW-08)

#### 3.2.4. Theme 3: Community-Driven Strategies for Health Promotion

The participants offered diverse recommendations for overcoming barriers and promoting HIVST as a sustainable, community-driven health strategy in the post-COVID-19 era.

##### Peer-to-Peer Support and Education

The participants emphasised that peer educators play a crucial role in demystifying HIVST, building trust, and encouraging hesitant individuals to adopt self-testing.


*“Peer-to-peer conversations are important because we can easily detect one who might be infected by noticing certain complaints like constant headaches or sweating. The strategy is to bring the person closer in a non-threatening way, talk to them as a peer, educate them about HIV self-testing, and encourage them to take control of their health.”*
(FSW-05)


*“Hearing about self-testing from someone they trust makes it easier for them to accept and act on the information without feeling judged or afraid.”*
(FSW-05)


*“Sometimes we listen to friends more than to health workers because friends know how to talk to us without judgment.”*
(FSW-14)

##### Integration into Public Health Systems and Insurance Schemes

The participants proposed incorporating HIVST into national health insurance programmes to enhance affordability and build public trust.


*“Providing free health schemes, like the National Health Insurance Scheme, will help raise awareness and increase uptake of HIV self-testing.”*
(FSW-10)


*“If HIV self-testing is included in government health programs, people will see it as something important and normal, not something secret or shameful.”*
(FSW-04)


*“It would also encourage more people to get tested regularly without worrying about financial barriers.”*
(FSW-13)

##### Community Awareness Campaigns and Practical Training

The participants suggested using widely accessed media platforms and public demonstrations to improve community knowledge and confidence in using HIVST.


*“They should create awareness through social media and radio stations because these are platforms that reach a lot of people easily. People often listen to the radio during their free time, whether at work or at home, and many of us are on social media every day.”*
(FSW-01)


*“You can educate people about it and do some training because many people might not know how to use the kit properly on their own. Without proper guidance, they might make mistakes or misinterpret the results.”*
(FSW-01)


*“If they show how to use it step by step, people will not be afraid to try it. People want to see how it works, not just hear about it.”*
(FSW-02)

The participants also praised the speed of HIVST, noting that rapid results allow timely decisions and reduce the stress associated with waiting for clinic-based testing.


*“Something happened where I used to stay, two people engaged in sexual intercourse, and by mistake, the condom burst during the act. In situations like that, having an HIV self-testing kit on hand is very important. It would allow people to act immediately, check their status on the spot, and reduce the panic and delays that come with trying to find a clinic or waiting in long lines.”*
(FSW-01)


*“It’s fast, and you receive your result immediately, which makes a big difference compared to going to a hospital.”*
(FSW-01)


*“At least you know immediately what your result is and can decide what to do next instead of waiting days for hospital results.”*
(FSW-04)

## 4. Discussion

This study examined HIVST as a strategy to empower communities and promote health in SSA, especially in the aftermath of the COVID-19 pandemic. Using an explanatory sequential mixed-methods design, the findings revealed low HIVST uptake (2.5%), with awareness at only 16.3%. Uptake was significantly associated with higher education (aOR, 7.36), urban residence, prior HIV testing, and better knowledge of HIV transmission. Qualitative insights highlighted privacy, autonomy, and convenience as key drivers of HIVST preference, while barriers included cost concerns, fear of handling results alone, and potential stigma. The participants proposed community-led strategies, including peer education and integration of HIVST into public health programmes, to improve uptake.

In our study, only 16.3% of the respondents had heard of HIV self-testing (HIVST), and just 2.5% had ever used it. These figures are substantially lower than those reported in other studies conducted in sub-Saharan Africa. For example, a cluster-randomised trial in Malawi observed that community-led HIVST achieved an uptake of 74.7% among the participants, and 42.5% following door-to-door distribution of kits [14]. In Kenya, awareness of oral HIVST was reported at 19% among young adults, with 75% expressing willingness to use HIVST in the future [23]. The lower uptake in our setting may reflect differences in programmatic investment, availability of community-led distribution models, and levels of community engagement and sensitisation across regions.

In our analysis, education emerged as a significant determinant of HIVST use, with individuals holding a tertiary education being 7.36 times more likely to use HIVST compared to those with no formal education. Although this adjusted odds ratio is notably higher than estimates from some prior studies in sub-Saharan Africa, our finding aligns with evidence from several recent investigations. For instance, an analysis of 21 countries across sub-Saharan Africa found that women with secondary or higher education had more than three times higher odds of HIVST knowledge and utilisation compared to those with no formal education (aOR, 3.08; 95% CI, 2.79–3.41) [24]. Similarly, among adults in Kenya, secondary education was associated with increased odds of HIVST use (aOR, 3.5; 95% CI, 2.1–5.9) [25]. Studies in university settings further support the importance of educational context: In Uganda, 19% of female university students reported having ever used HIVST, with educational environment facilitating both awareness and acceptability [26]. Likewise, research among tertiary-level students in South Africa and Zimbabwe highlighted substantial interest in HIVST, though prior utilisation remained low in some groups [27,28]. The evidence collectively indicates that higher educational attainment is an important enabler of HIVST uptake. These findings underscore the potential for educational institutions to serve as effective platforms for scaling up HIVST interventions.

Wealth status was significant in our analysis, with individuals in the richest quintile being 3.28 times more likely to use HIVST than those in the poorest group. This aligns with evidence from sub-Saharan Africa, where wealthier individuals consistently show higher HIV testing and HIVST uptake. For example, women from the richest households in Ghana had over four times higher odds of HIVST use (aOR, 4.31) than those from the poorest households [29]. Studies in South Africa also found higher HIVST knowledge among wealthier women [30]. These patterns underscore persistent socioeconomic disparities in HIV testing uptake.

While urban residence was associated with increased HIVST use in our study, previous research has shown mixed results, with some studies reporting higher uptake in urban settings due to better access to health commodities [24,31], while others have documented persistent barriers related to stigma even in urban environments [32].

Knowledge of HIV transmission emerged as a strong predictor of HIVST uptake in our analysis, with an adjusted odds ratio of 33.43. Evidence from prior studies has consistently linked HIV knowledge to increased HIVST use. For instance, in Ghana, women with higher education, a proxy for HIV knowledge, had substantially higher odds of HIVST uptake compared to those with no education [29]. Similarly, awareness of sexually transmitted infections was strongly associated with HIVST use among women in 21 sub-Saharan African countries (aOR, 7.47) [24].

Prior HIV testing was also a significant predictor (aOR, 3.33), consistent with findings from [33], who reported that individuals previously tested for HIV were more likely to adopt HIVST. Condom use at last sexual intercourse was associated with a modest increase in HIVST uptake (aOR, 1.49), echoing findings from other studies where safer sexual practices correlated with proactive health behaviours [34].

The qualitative findings in this study highlighted privacy and autonomy as primary motivators for HIVST uptake. These themes are consistent with research conducted among key populations in SSA, where privacy has been documented as a significant advantage of HIVST. Studies in Côte d’Ivoire, Mali, Senegal, and Benin have similarly reported that privacy, convenience, and autonomy are major drivers of HIVST preference among female sex workers and other key populations [35,36].

Cost emerged as a major barrier in our study, with the participants expressing concerns that even moderately priced HIVST kits remain unaffordable for those living in poverty. Evidence from Eswatini shows that community and workplace HIVST distribution averaged USD 17.23 per kit distributed and USD 18.91 per person tested [37]. Systematic reviews across low- and middle-income countries indicate costs per person tested ranging from as low as USD 1.09 up to USD 155, depending on the programme design and support services [38]. In Kenya, integration of HIVST into assisted partner services costs approximately USD 8.97 per kit [39]. Despite these relatively low prices compared to facility-based testing in some contexts, willingness to pay among end-users often drops sharply if kit prices exceed USD 2–3, highlighting the affordability gap for low-income populations [38]. These findings emphasise that for HIVST to be scalable and equitable, costs must be further reduced or subsidised, especially for populations most affected by HIV.

Fear of testing alone and handling a positive result independently emerged as significant barriers in our qualitative analysis. This mirrors findings from studies in Zambia and Malawi, where apprehension about emotional distress and lack of immediate counselling has deterred individuals from using HIVST [40,41]. Furthermore, concerns about test accuracy have been widely reported, including among sex workers in Tanzania, where doubts regarding reliability significantly reduced uptake [42].

The importance of peer networks for promoting HIVST was prominent in our qualitative findings. The participants emphasised the role of trusted peers in building confidence and overcoming stigma. This observation is consistent with findings from a global systematic review and network meta-analysis where peer-led distribution of HIVST kits resulted in significantly higher uptake and identification of new HIV infections compared to standard facility-based testing [43]. Peer-driven strategies have been shown to be particularly effective in reaching marginalised populations and sustaining engagement in HIV prevention services.

The participants in this study advocated for integrating HIVST into national health schemes to improve affordability and access. This recommendation aligns with findings from other SSA contexts, where incorporation of HIVST into public health programmes has been proposed as essential for scale-up and sustainability [7,44]. Integration not only reduces cost barriers but also contributes to normalising HIVST as a routine component of HIV prevention efforts.

### 4.1. Triangulation of the Quantitative and Qualitative Findings

The explanatory sequential mixed-methods design employed in this study allowed for a comprehensive and layered understanding of HIVST dynamics in SSA. The quantitative findings demonstrated that HIVST uptake was significantly associated with education [24,45], wealth [46], prior HIV testing [33], and HIV knowledge, reflecting enduring structural inequalities across the region, consistent with these multi-country studies that attribute testing disparities to socioeconomic status and educational gradients.

The qualitative findings expanded upon these statistical relationships by revealing the psychosocial and contextual mechanisms underlying HIVST decisions. For example, participants with higher education described feeling more competent in interpreting results and making subsequent health decisions, insights that parallel qualitative work in SSA, where health literacy and self-efficacy emerged as strong enablers of HIVST [47]. Similarly, the quantitative association between wealth and HIVST uptake was contextualised by narratives around affordability, opportunity costs, and limited subsidy mechanisms, echoing findings from a scoping review where cost perceptions remained a persistent deterrent [7].

Furthermore, while quantitative data confirmed that urban residence and HIV knowledge predicted higher HIVST use, the qualitative narratives illustrated why: Urban participants benefited from informal peer networks, trusted intermediaries, and access to HIVST commodities through pharmacies or outreach programmes. The emphasis on privacy and autonomy observed in our qualitative data also resonates with global HIVST studies among key populations [7,47,48,49], highlighting that confidentiality and self-agency are central motivators across contexts.

It is important to note that the quantitative and qualitative components of this study involved different populations and temporal contexts. The quantitative data, drawn from the Demographic and Health Surveys (2015–2022), represented men and women across 24 sub-Saharan African countries, while the qualitative narratives were collected in 2023 among FSWs in Nigeria. These populations were therefore not intended to be directly compared or statistically merged. Rather, the integration was interpretive, with each component serving a distinct but complementary function. The DHS analysis provided macro-level evidence of regional patterns and disparities in HIV testing and self-testing, whereas the qualitative inquiry offered micro-level perspectives on how a highly vulnerable subpopulation experienced HIV self-testing in the post-COVID-19 era. This form of cross-level interpretive integration, consistent with the guidance of Creswell and Clark [20] and Fetters, Curry, and Creswell [21], links regional trends to lived community realities and enhances the relevance of the findings for policy and programme design.

### 4.2. Implications for Policy and Practice

This study highlights key priorities for HIV prevention in sub-Saharan Africa as health systems adapt post-COVID-19. Affordability remains essential, requiring subsidies and potential inclusion of HIV self-testing (HIVST) in national health insurance to ensure access for low-income groups. Community engagement through peer-led outreach can reduce stigma and boost trust in HIVST. Robust education and support efforts are needed to dispel misconceptions, guide proper use, and promote timely care linkage. Finally, integrating HIVST into existing health systems is critical for sustainable supply chains and alignment with routine services. Collectively, these strategies are vital for sustaining HIV prevention gains and building resilient, community-centred care.

### 4.3. Strengths and Limitations

A key strength of this study lies in its explanatory sequential mixed-methods design, which enabled interpretive integration of quantitative and qualitative evidence across different contexts and levels of analysis. The large, multi-country DHS sample (2015–2022) enhances the generalisability of the quantitative findings, while the 2023 qualitative narratives from female sex workers in Nigeria provide contextual and experiential depth to those regional patterns. This design allowed the qualitative component to expand and explain the quantitative trends rather than statistically merge the two datasets, thereby linking macro-level structural determinants with micro-level lived realities.

However, certain limitations should be acknowledged. The quantitative data were derived from self-reported survey responses, which are subject to recall and social desirability bias. The qualitative component, conducted among a single key population group and within one urban setting, may limit transferability to other sub-populations or rural contexts. Additionally, the study did not distinguish between different HIVST modalities (oral fluid vs. blood-based), which could influence acceptability and perceptions. Despite these constraints, the study’s interpretive, multi-level approach provides valuable insights for designing equitable and context-sensitive HIVST interventions in sub-Saharan Africa.

Recruitment was facilitated through collaboration with Youth PRO-FILE, a local non-governmental organisation with extensive experience in HIV prevention among FSWs. Because most participants were drawn from brothels within the Port Harcourt metropolis, the resulting sample may modestly overrepresent younger participants and religious and relationship status distribution, reflecting the demographic composition of brothels within this locality. While this limits generalisability, the qualitative component was designed to generate in-depth experiential insights rather than statistically representative estimates, consistent with interpretive mixed-methods principles.

### 4.4. Implications for Future Research

Future research should examine differences in acceptability and uptake between various HIVST modalities and supervision models (e.g., supervised vs. unsupervised testing). Longitudinal studies are needed to assess sustained HIVST usage, linkage-to-care outcomes, and the public health impact of integrating HIVST into national HIV prevention programmes. Further exploration of digital platforms and innovative peer-led interventions could offer insights into scalable approaches for expanding HIVST coverage across diverse populations in SSA.

## 5. Conclusions

This study confirms that HIVST is a promising tool for empowering communities and enhancing HIV prevention efforts in SSA, particularly in the post-COVID-19 era. While HIVST offers significant benefits in terms of privacy, autonomy, and convenience, addressing persistent barriers related to cost, stigma, and knowledge gaps is crucial for achieving widespread and equitable adoption. Implementing community-driven strategies, robust policy frameworks, and targeted educational interventions will be essential for transforming HIVST into a sustainable, widely adopted public health intervention across the region.

## Figures and Tables

**Figure 1 ijerph-22-01616-f001:**
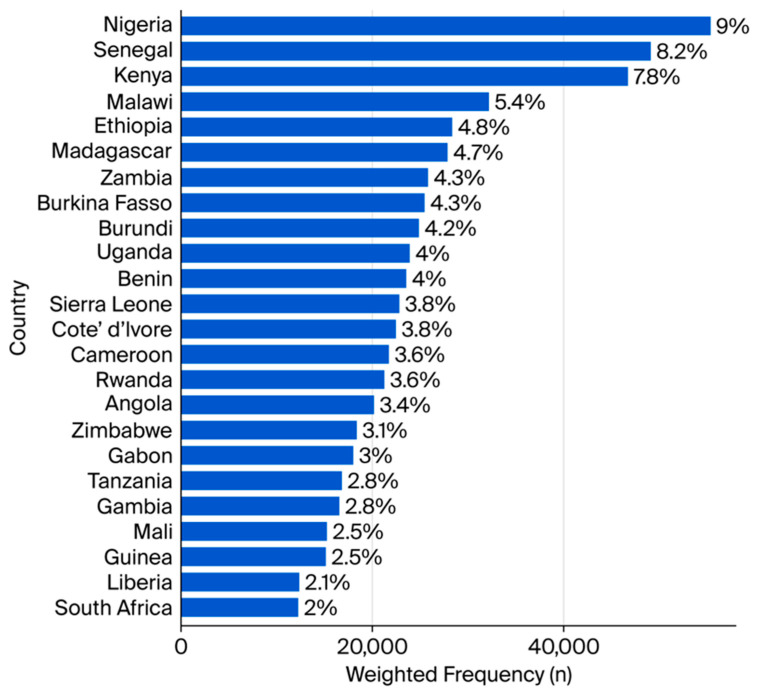
Weighted Frequency and Percent by Country.

**Table 1 ijerph-22-01616-t001:** Sociodemographic characteristics of the respondents (n = 594,639).

Variables	Unweighted Frequency (n)	Weighted Frequency (n)	Weighted Percentage (%)
**Current age**			
Mean ± SD		29.43 ± 10.48 ^∞^	
Range		(15–64)	
**Gender**			
Male	185,411	185,411.2	31.2
Female	409,228	409,228.0	68.8
**Current marital status**			
Never in union	198,070	199,780.7	33.6
Married	294,074	292,209.7	49.1
Living with a partner	60,424	60,508.35	10.2
Widowed	11,242	10,850.9	1.8
Divorced	10,883	10,849.5	1.8
No longer living together/separated	19,946	20,439.9	3.4
**Currently residing with husband/partner** (n = 352,714)			
Yes	303,840	304,189.3	86.2
No	50,656	48,525.6	13.8
**Type of place of residence**			
Urban	230,854	247,302.3	41.6
Rural	363,785	347,336.9	58.4
**Highest educational level**			
No education	165,963	158,109.6	26.6
Primary	184,162	182,851.2	30.8
Secondary	204,768	208,284.4	35.0
Higher	39,743	45,389.9	7.6
**Wealth index combined** (n = 594,639)			
Poorest	120,696	101,738.9	17.1
Poorer	113,720	109,177.8	18.4
Middle	117,999	115,520.8	19.4
Richer	117,448	126,987.3	21.4
Richest	124,776	141,214.4	23.7

^∞^ Mean ± standard deviation.

**Table 2 ijerph-22-01616-t002:** Awareness, Attitude and uptake of HIV self-testing (n = 594,639).

Variables	Unweighted Frequency (n)	Weighted Frequency (n)	Weighted Percentage (%)
**Good knowledge of HIV/AIDS transmission** (n = 567,880.494)			
Yes	482,270	485,014.1	93.7
No	36,076	32,870.1	6.3
**Condom used during last sex with most recent partner** (n = 424,130)			
Yes	49,585	50,753.9	12.0
No	373,816	373,376.9	88.0
**Ever been tested for HIV** (n = 498,638)			
Yes	254,839	256,924.4	51.5
No	243,984	241,714.1	48.5
**Know a place to get HIV test** (n = 369,284)			
Yes	302,192	303,118.3	82.1
No	64,693	66,166.4	17.9
**Received result from last HIV test** (n = 256,924)			
Yes	244,801	246,785.6	96.1
No	10,038	10,138.9	3.9
**Fear of stigma** (n = 392,989)			
Yes	30,8516	312,261.7	79.5
No	61,741	60,740.5	15.5
Don’t know/not sure/depends	20,529	19,987.2	5.1
**Ever heard of HIVST** (n = 395,614)			
Yes	61,127	64,344.7	16.3
No	33,2671	33,1270.1	83.7
**Ever used HIVST** (n = 395,614)			
Yes	9955	9955.6	2.5
No	385,659	385,659.2	97.5

**Table 3 ijerph-22-01616-t003:** Sociodemographic and economic determinants of HIV self-testing uptake (n = 594,639).

Variables	cOR (95% CI)	*p*-Value	aOR (95% CI)	*p*-Value
**Gender**				
Male	1.06 (1.01–1.11)	0.009 *	0.89 (0.84–0.94)	0.001 *
Female ^R^	-	-	-	-
**Marital status**				
Never in union ^R^	-	-	-	-
Married	0.57 (0.52–0.63)	<0.001 *	1.95 (1.78–2.15)	0.001 *
Living with partner	0.65 (0.60–0.71)	<0.001 *	1.35 (1.23–1.47)	0.001 *
Widowed	0.51 (0.46–0.57)	<0.001 *	1.86 (1.67–2.08)	0.001 *
Divorced	0.39 (0.32–0.48)	<0.001 *	1.70 (1.38–2.11)	0.001 *
**Currently residing with Partner**				
Yes	0.55 (0.52–0.59)	<0.001 *	1.58 (1.48–1.68)	0.001 *
No ^R^	-	-	-	-
**Place of residence**				
Urban	2.21 (2.12–2.30)	<0.001 *	0.92 (0.86–0.98)	0.006 *
Rural ^R^	-	-	-	-
**Highest educational level**				
No education ^R^	-	-	-	-
Primary	0.08 (0.08–0.09)	<0.001 *	2.36 (2.19–2.55)	0.001 *
Secondary	0.12 (0.11–0.12)	<0.001 *	4.70 (4.28–5.17)	0.001 *
Higher	0.26 (0.24–0.27)	<0.001 *	7.36 (6.62–8.18)	0.001 *
**Wealth index**				
Poorest ^R^	-	-	-	-
Poorer	0.26 (0.25–0.29)	<0.001 *	1.66 (1.54–1.79)	<0.001 *
Middle	0.35 (0.33–0.38)	<0.001 *	2.46 (2.24–2.70)	<0.001 *
Richer	0.39 (0.37–0.42)	<0.001 *	2.65 (2.41–2.92)	<0.001 *
Richest	0.62 (0.58–0.65)	<0.001 *	3.28 (2.95–3.65)	<0.001 *

* Statistically significant (*p* ≤ 0.05); ^R^, reference; cOR, crude odds ratio (unadjusted); aOR, adjusted odds ratio.

**Table 4 ijerph-22-01616-t004:** Knowledge, behavioural, and psychosocial determinants of HIV self-testing uptake (n = 594,639).

Variables	cOR (95% CI)	*p*-Value	aOR (95% CI)	*p*-Value
**Good knowledge of HIV/AIDS transmission**				
Yes	54.02 (23.51–124.14)	<0.001 *	33.43 (11.03–101.24)	<0.001 *
No ^R^	-	-		
**Condom used (last sex)**				
Yes	1.56 (1.47–1.65)	<0.001 *	1.49 (1.15–1.93)	0.002 *
No ^R^	-	-	-	-
**Ever tested for HIV**				
Yes	5.65 (5.32–6.01)	<0.001 *	3.33 (3.08–3.60)	<0.001 *
No ^R^	-	-	-	-
**Know test location**				
Yes	3.99 (3.59–4.44)	<0.001 *	1.52 (1.33–1.72)	<0.001 *
No ^R^	-	-	-	-
**Received last test result**				
Yes	2.83 (2.37–3.38)	<0.001 *	2.22 (1.84–2.68)	<0.001 *
No ^R^	-	-	-	-
**Fear of stigma**				
Yes	2.92 (2.46–3.48)	<0.001 *	0.49 (0.41–0.59)	<0.001 *
No	2.05 (1.70–2.47)	<0.001 *	0.34 (0.29–0.41)	<0.001 *

* Statistically significant (*p* ≤ 0.05); ^R^, reference; cOR, crude odds ratio (unadjusted); aOR, adjusted odds ratio.

**Table 5 ijerph-22-01616-t005:** Sociodemographic characteristics of the respondents (n = 15).

Variables	Frequency (n)	Percentage (%)
**Age**		
18–30	12	80.0
31–40	3	20.0
Mean ± SD	29.33 ± 4.2 ^∞^	
Range	(23–38)	
**Religion**		
Christian	11	73.3
Muslim	1	6.7
No religious affiliation	3	20.0
**Relationship status**		
Single	10	66.7
Living with partner	1	6.7
I choose not to answer	4	26.7
**Have children**		
Yes	1	6.7
No	11	73.3
I choose not to answer	3	20.00
**Highest level of education**		
Secondary school (high school)	7	46.7
Completed OND	2	13.3
Completed Bachelor’s degree	6	40.0
**Engaged in any other occupation**		
Yes	8	53.3
No	7	46.7
**Nature of occupation (n = 8)**		
Employed or self-employed full-time work	1	12.5
Employed or self-employed part-time work	7	87.5

^∞^ Mean ± standard deviation.

**Table 6 ijerph-22-01616-t006:** Themes and sub-themes emerging from the participants’ perspectives on HIV self-testing for community empowerment in post-COVID-19 sub-Saharan Africa (n = 15).

Themes	Sub-Themes
Empowerment through HIVST	- Privacy and Confidentiality
	- Autonomy and Personal Control
	- Convenience and Rapid Results
Barriers and Vulnerabilities Post-COVID-19	- Financial Barriers and Affordability Concerns
	- Fear, Anxiety, and Emotional Distress
	- Perceived Stigma and Confidentiality Risks
Community-Driven Strategies for Health Promotion	- Peer-to-Peer Support and Education
	- Integration of HIVST into Public Health Systems and Insurance Schemes
	- Community Awareness Campaigns and Practical Training

## Data Availability

The data supporting the findings of this study are available through public repositories. The quantitative dataset derived from the Demographic and Health Surveys (DHSs, 2015–2022) and used for the HIV self-testing analysis is accessible at https://doi.org/10.6084/m9.figshare.30281185 (created on 5 October 2025). The qualitative transcripts generated from interviews with female sex workers in Port Harcourt, Nigeria (May 2023), are available at https://doi.org/10.6084/m9.figshare.30281197 (created on 5 October 2025). Due to confidentiality and ethical considerations, identifiable portions of the qualitative data are restricted.

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
