# Peer review of "Empowering Vulnerable Communities Through HIV Self-Testing: Post-COVID-19 Strategies for Health Promotion in Sub-Saharan Africa"

_ijerph, 2025, doi:10.3390/ijerph22111616_

Round 1
Reviewer 1 Report
Comments and Suggestions for Authors
Estimated Authors,
I've read with great interest the present study from Sibiya et al. providing a valuable hindsight on the topic of HIV self-testing from Sub-Saharian Africa. According to this study, a significant share of population may be affected by low awareness and low uptake, a results that in fact may impair any strategy aimed to control and manage currently still ongoing HIV pandemic.
The paper benefits from a quantitative and qualitative design, and both sections are properly designed and reported, but some suggestions may be envisaged for further improving the overall quality of the paper.
1) To begin with, it is quite unclear how participants were selected. Section 2.2 provides some information about this topic, but as the study focuses on a specific topic (i.e. HIV-ST), which may be affected by specific personal and socioeconomic features, Authors should properly address the selection of participants in order to rule out any potential self-selection of participants, and/or the oversampling of certain population groups. this is particularly significant when dealing with refusal to participate into a certain study.
2) participants were from various geographical areas, but was the population proportionally established (on the basis of residing population of parent countries)? please explain
3) Table 1 must be improved by crude frequency
4) The same for Table 2
5) Authors should address in comments how the sampling strategy may impair the general representativity of the qualitative section (i.e. oversampling of a certain age group, and even the religious affiliation of most participants suggests the potential oversampling of a certain ethnic group compared to other ones)
Author Response
We appreciate the reviewers’ constructive feedback, which has significantly improved our manuscript. We hope the revised version meets the journal’s standards and look forward to your favorable consideration.
Please see attachment for response.

Reviewer 2 Report
Comments and Suggestions for Authors
This manuscript describes socio-demographic and economic characteristics of the uptake of HIV self-testing (HIVST) in 24 sub-Saharan countries using DHS data sets of surveys conducted between 2009 and 2019. It then also goes on to describe themes that emerged from interviews with FSW in Nigeria when discussing their understanding and experiences with HIVST.
Introduction: adequately discusses the importance of the need for additional strategies to facilitate expansion of HIV testing using strategies such as HIVST. Appropriately discusses how the COVID-19 pandemic impacted HIV testing and therefore, HIV epidemic control.
Methods: Unclear that we can truly merge these data sets together, given 1) quantitative data is pre-pandemic, so it really does not answer the question of the socio-demographic variables associated with HIVST during or after the pandemic, 2) the populations used in the QUANT analysis and the QUAL analysis are 2 very different populations. One is a mix of men and women across sub-Saharan Africa; the other is FSW in Nigeria. This makes it difficult to truly triangulate and merge the data types, as they were both collected at different points in relation to the pandemic, and also are of very different populations.
Unclear when the FSW interviewed. Was this post-COVID or during the pandemic?
Results section: Need to format/correct tables: Both Table 1 and Table 2 report the “wealth index”; would round up reported numbers to whole numbers, not 3 decimal places.
Duplicative reporting of data in both written form and in tables. I.e., no need to describe data in written form if already in a table. E.g. is Table 5 data is mostly also written out in the results section ( lines 278-283)
Discussion and Conclusions: Adequately discuss the individual qualitative and quantitative data in relation to existing data. Very little discussion on merging the data for a paper that emphasizes a mixed methods design.
Author Response

(The authors gave the same response as above.)

Round 2
Reviewer 1 Report
Comments and Suggestions for Authors
Estimated Authors,
the paper has been sufficiently improved and therefore I'm endorsing its acceptance
Reviewer 2 Report
Comments and Suggestions for Authors
This manuscript uses an explanatory sequential mixed methods design to understand socio-demographic and economic characteristics of HIV self-testing (HIVST) using DHS data sets of surveys from 24 SSA countries taken between 2015-2022 and interviews with FSW to help inform strategies to expand the use of HIVST post-COVID-19.
The authors have sufficiently addressed prior major concerns and clarified their methodological approach. The clarification of the years of data collection and data integration methods is much clearer.
Minor other general notes:
To make the paper more concise and readable, consider reducing its overall word count. Examples of places where word count reduction could be easily achieved include
Limiting the number of representative participant quotes used under the qualitative theme to 1-2 per sub-theme
Lines 130-137, which describe the aim of the paper, could also be condensed into a sentence, especially since the line above introduces what the paper aims to do.
